# Antisense and Gene Therapy Options for Duchenne Muscular Dystrophy Arising from Mutations in the N-Terminal Hotspot

**DOI:** 10.3390/genes13020257

**Published:** 2022-01-28

**Authors:** Harry Wilton-Clark, Toshifumi Yokota

**Affiliations:** Department of Medical Genetics, Faculty of Medicine and Dentistry, University of Alberta, Edmonton, AB T6G 2R3, Canada; hwiltonc@ualberta.ca

**Keywords:** Duchenne muscular dystrophy, exon skipping therapy, antisense oligonucleotide, microdystrophin, adeno-associated virus

## Abstract

Duchenne muscular dystrophy (DMD) is a fatal genetic disease affecting children that is caused by a mutation in the gene encoding for dystrophin. In the absence of functional dystrophin, patients experience progressive muscle deterioration, leaving them wheelchair-bound by age 12 and with few patients surviving beyond their third decade of life as the disease advances and causes cardiac and respiratory difficulties. In recent years, an increasing number of antisense and gene therapies have been studied for the treatment of muscular dystrophy; however, few of these therapies focus on treating mutations arising in the N-terminal encoding region of the dystrophin gene. This review summarizes the current state of development of N-terminal antisense and gene therapies for DMD, mainly focusing on exon-skipping therapy for duplications and deletions, as well as microdystrophin therapy.

## 1. Introduction

Duchenne muscular dystrophy (DMD) is a fatal X-linked recessive disease caused by an inability to produce functional dystrophin, a protein critical for muscular strength and stability, and the absence of which results in progressive muscular deterioration and fibrosis [1,2,3]. Most DMD patients present with significant lower limb weakness and resulting ambulatory difficulty before the age of 5, progressing to the point of mandatory wheelchair use by age 12 [4]. By their late teenage years, most patients suffer from cardiac and respiratory impairment, indicating disease progression to cardiac muscles and the diaphragm. As respiratory difficulties worsen, ventilatory support becomes necessary for survival. DMD affects approximately 1/3500–1/5000 male births globally, and most patients do not survive beyond their third decade of life [1,5].

Dystrophin is encoded by *DMD*, a 1.2 million base-pair gene containing 79 exons and located at the Xp21 locus [6,7,8]. Approximately 60–70% of the mutations resulting in DMD are whole-exon deletions or duplications of *DMD* that distort the open-reading frame (ORF) of the gene, preventing the production of a functional protein product [9,10]. The remainder of DMD-causing mutations are typically small nonsense mutations; however, intronic, splice site, and 5′ and 3′ untranslated region (UTR) mutations have also been reported. Large deletions do not distribute evenly across the *DMD* gene, and previous studies have identified mutation hotspots at exons 2–22 and exons 43–55, which contain 23% and 73% of major deletions, respectively [11].

No cure is currently available for DMD, and most existing standards of care aim to delay disease progression through corticosteroid treatment, followed by end-stage ventilatory and cardiac support when the need arises [12,13]. While this approach is often able to extend the life of DMD patients into their mid-to-late twenties, it is far from curative. Furthermore, corticosteroid treatment is associated with various side effects, including unintentional weight gain, delayed growth, delayed onset of puberty, bone weakness, and adrenal insufficiency [14]. In recent years, the suitability of gene and antisense therapies to treat DMD have been explored in a variety of approaches, and four drug candidates have received FDA approval and are now available on the market to treat a certain subset of DMD patients [15,16,17,18]. Notably, all four currently approved drugs treat patients with mutations in the exon 43–55 hotspot, while nothing is available yet for patients with mutations in the N-terminal-encoding exon 2–22 hotspot. The purpose of this article is to highlight the current state of development for precision therapies focusing on the currently underserved exon 2–22 hotspot.

Precision therapies for DMD include gene therapy and antisense therapy. Gene therapy tends to be used as an umbrella term for a variety of more specific approaches, and while the exact definition varies widely, it usually refers to therapies that rely on the production of recombinant genetic materials, such as microdystrophin and CRISPR/Cas9, provided to cells [19,20,21]. Gene therapy has the potential to provide long-lasting therapy with a single treatment, effectively treating some diseases at their root. However, unwanted mutations, immunogenicity, and off-target effects can pose a safety risk to patients, limiting the applicability of some methods of gene therapy [21]. In comparison, antisense therapy directly treats patients with synthetically produced DNA-like molecules known as antisense oligonucleotides (AONs) targeted for mRNA without providing new genes to express, in the same fashion as many other drugs might be provided to a patient [22,23]. While AONs generally have a favorable safety profile, the requirement for repeated treatments can add cost and complexity, especially in the setting of chronic disease [16].

## 2. Exon 2 Skipping for Exon 2 Duplication—Astellas Gene Therapies

Astellas Gene Therapies (formerly Audentes Therapeutics) is a clinical-stage gene-therapy company aiming to develop treatments for a variety of rare genetic disorders, such as X-linked myotonic myopathy, Pompe disease, and DMD. Astellas specializes in adeno-associated virus (AAV)-related technologies, and most of its gene therapies rely on various AAV delivery vectors. One of its lead candidates, scAAV9.U7.ACCA, is currently in phase-1/2 clinical trials and aims to treat DMD patients with a duplication of exon 2.

AAVs are a class of viruses in the Parvovirus family that can infect both dividing and non-dividing cells in humans, but they are not believed to cause disease [24,25,26]. This lack of pathogenicity, combined with a relatively small genome of 4.8 kilobases, has made AAV an attractive option for viral gene therapy, and hundreds of AAV-based therapies have undergone or are currently undergoing clinical trials [24,27,28]. AAV is a simple structure containing the viral ssDNA genome enclosed in a protein capsid, which plays important roles in protection, cell membrane permeation, and cellular targeting [29]. In its endogenous state and upon entering a host cell, AAV would either form a concatemer to exist in an extrachromosomal fashion, or integrate at the AAVS1 site located on chromosome 19 of the human genome. In order to favor the former pathway, recombinant variants of AAV harbor deletions of certain proteins necessary for integration, preventing the ability of recombinant AAV to integrate at the AAVS1 site. Thus, the therapeutic provision of AAV takes advantage of native viral capsid targeting to introduce an extrachromosomal concatemer, which remains in target cells for approximately 6–12 months, while continuously expressing the recombinant gene(s) of interest [30]. Previous studies have suggested that the reliance on the viral host to synthesize a second strand from viral ssDNA is often a bottleneck in therapeutic efficiency [31]. To address this issue, many groups use self-complementary ssDNA vectors known as scAAV, which can fold and base pair with themselves, expediting the production of the recombinant therapeutic gene.

In the case of Astellas, the recombinant genes of interest are four non-coding U7 small nuclear RNA (U7snRNA) molecules modified to target and bind splice acceptor and donor sites flanking exon 2 of *DMD* [32]. These snRNAs are designed to sterically prevent the binding of other splicing enzymes on the pre-mRNA transcript, resulting in the exclusion of the flanked exon from the final mature mRNA transcript. By delivering these molecules in a scAAV9 vector to patients with a duplication of exon 2, Astellas aims to induce the skipping of one of the exon 2 duplicates, restoring the production of regular full-length dystrophin. Astellas has previously demonstrated that the treatment of mice containing a duplication of exon 2 with scAAV9.U7.ACCA can induce skipping of exon 2 and result in amelioration of the DMD phenotype [33]. As the U7snRNA is unable to differentiate between the identical duplicates of exon 2, there exists the possibility of skipping both copies. However, Astellas found that, even when there is complete exclusion of both copies of exon 2, production of a truncated but functional protein is maintained through alternative translation initiation at an internal ribosome entry site (IRES) on exon 5 in both human cells and mice [34]. This finding suggests that the therapeutic window for exon skipping in exon-2-duplication patients is larger than previously expected, and it indicates that no additional measures are required to favor single-exon skipping.

Preclinical safety studies in mice have explored the off-target activity of scAAV9.U7.ACCA by sequencing the transcriptome from skeletal muscle, liver, and heart tissues after 90 days of intramuscular treatment [32,35]. Local splice variation (LSV) analysis in these tissues identified 16 LSVs, only one of which was present in both skeletal and cardiac tissue, and the majority of which were previously documented as known events in the Vertebrate Alternative Splicing and Transcription Database (VastDB) [32,35]. Based on these findings, the off-target activity of scAAV9.U7.ACCA was determined to be low. To further prepare for clinical trials, Astellas assessed the toxicity of scAAV9.U7.ACCA in nonhuman primates [36]. Male juvenile cynomolgus monkeys were treated with either 3 × 10^13^ vg/kg or 8 × 10^13^ vg/kg of scAAV9.U7.ACCA and assessed for toxicity at 3-month and 6-month time points. Toxicity tests included clinical assessments, biochemical and hematology assays, and histological analysis. No significant toxicity was observed at either dose, and mRNA sequencing demonstrated significant skipping of *DMD* exon 2 which occurred in a dose-dependent manner [36]. Based on these promising preclinical findings, Astellas began a phase-I/II clinical trial for scAAV9.U7.ACCA in early 2020 (NCT04240314) [37].

NCT04240314, officially titled “Phase I/IIa Systemic Gene Delivery Clinical Trial of scAAV9.U7.ACCA for Exon 2 Duplication-Associated Duchenne Muscular Dystrophy”, is an open-label study primarily designed to assess the safety of scAAV9.U7.ACCA in DMD patients with a confirmed *DMD* exon-2 duplication. Secondary outcomes will also explore preliminary efficacy data. Three participants were treated with the minimal efficacious dose of scAAV9.U7.ACCA via peripheral limb vein injection. Inclusion criteria for the study required all participants to be males between the ages of 6 months and 14 years with a confirmed duplication of exon 2 in the *DMD* gene and who are otherwise healthy with a confirmed absence of AAV9 antibodies. All participants will be monitored for adverse effects over the course of 2 years, observing for toxicities as defined by Common Terminology Criteria for Adverse Events (CTCAE) 5.0 [38]. Dystrophin expression will be assessed over the course of 1 year, with Western blotting and immunofluorescent staining of muscle biopsies, and compared to data from baseline pretreatment biopsies. Exclusion of exon 2 from the final DMD mRNA transcript will also be quantified with RT-PCR. These data will provide critical information about the safety and efficacy of scAAV9.U7.ACCA, informing the continuation or discontinuation of further clinical trials. The study is currently estimated to be completed by November 2025.

### Strengths and Weaknesses

AAV offers several salient advantages as a gene-therapy vector. It has low pathogenicity, has been relatively well studied, and the long-acting duration of concatemerized viral payload can help to reduce both the time and cost required for successful treatment, which is particularly important when compared to the high cost of other gene therapy options [16,39]. There are also a multitude of AAV serotypes available with different tissue targeting capabilities, and certain serotypes are able to effectively penetrate the blood–brain barrier, increasing its utility in treating CNS diseases, which can be difficult to target with other drugs [39].

However, AAV therapy is not without disadvantages. The small size of the virus imposes restrictions on which recombinant genes can be used, up to a maximum of 4.7 kb [25,40]. This limitation means that approaches requiring a larger gene insert, such as the production of recombinant full-size dystrophin for DMD, are impossible with AAV therapy [41]. The potential presence of AAV antibodies in prospective patients further reduces the applicability of AAV therapy [42]. Approximately 30–60% of the population test positive for anti-AAV antibodies due to a previous infection, and the presence of these antibodies could result in a host immune reaction and corresponding neutralization of the AAV vector, preventing therapeutic effect [42,43,44]. Even in a patient without natural infection, the first round of AAV treatment might stimulate the host humoral response and contribute to anti-AAV antibody development, reducing the efficacy of future AAV treatments.

More importantly than the loss of efficacy, this immune reaction also presents a safety risk to patients. Despite the typically harmless nature of AAV, serious treatment-related adverse effects, such as hepatotoxicity, myocarditis, microangiopathy, and nerve toxicity, have been observed in previous AAV-based clinical trials [45,46]. In 2020, three patients died following decompensated liver failure during a phase I/II trial for AAV-mediated treatment of X-linked myotubular myopathy, prompting the FDA to impose a clinical hold on the trial [47,48,49]. The study was resumed in 2021, and despite using a lower dose predicted to be safe, a fourth patient fatality occurred in September 2021, once again resulting in a clinical hold and raising serious concerns about the safety of AAV therapy [50,51]. With so many AAV therapies currently undergoing clinical trials, further validation of the safety of AAV must be of the utmost priority to ensure patient health.

To address the issue of AAV reactivity, recent research has focused on the development of novel AAV serovariants that are unrecognizable by common anti-AAV antibodies, which are generally anti-AAV2, and less likely to stimulate an immune response of their own [52]. Other approaches have explored the development of various adjuvants designed to prevent an immune response to AAV; however, providing immunosuppressants to a population with significant health complications brings its own set of additional risks [44,52,53]. Finally, as with all viral therapies, AAV poses the risk of off-target genomic integration and possible tumorigenesis [54]. Future studies, and the vast number of AAV-based candidates currently under clinical trial, will ultimately determine the extent to which AAV therapy poses a serious safety risk.

## 3. State of EST for Deletions

As previously mentioned, there are currently four antisense-mediated therapies available on the market which have received FDA approval to treat DMD. These therapies use an approach known as exon-skipping therapy (EST), which is similar but not identical to the exon skipping approach developed by Astellas to treat exon-2 duplications. Rather than treating patients with exon duplications, existing EST candidates treat patients whose DMD is attributable to whole-exon deletions in the *DMD* gene causing frameshift mutations.

EST uses AONs, which hybridize to the dystrophin pre-mRNA at strategically designed splice motif sites on either site of the target exon, blocking local spliceosome binding and excluding the flanked exon from the final mRNA transcript [55,56]. By excluding one or more exons adjacent to the mutation site, the reading frame downstream of the original deletion is restored, resulting in the production of internally deleted but functional dystrophin proteins (Figure 1) [57].

These truncated proteins have clinically significant benefits for patients and are expected to result in a phenotype resemblant of the much milder Becker muscular dystrophy (BMD) [58]. While this is similar to the mechanism of Astellas’ U7snRNA molecules, there are a few key differences between the two approaches. Primarily, the desired outcome of the U7snRNA treatment for exon-2 duplications is a regular full-length dystrophin transcript, whereas the desired outcome of EST for deletions is a truncated but functional dystrophin transcript. Furthermore, the chemistries of the drugs themselves differ greatly; EST AONs are intravenously injected synthetic phosphorodiamidate morpholino oligomers (PMOs), while the U7snRNA produced by concatemerized scAAV9 has a standard RNA structure. PMOs have a phosphorodiamidate backbone and six-sided morpholine ring, differing from the endogenously occurring phosphodiester backbone and five-sided ribose ring present in RNA [56,59]. Compared to RNA, these alterations provide PMOs with increased resistance to enzymatic degradation and a lower likelihood of interacting with natural proteins. Lastly, while scAAV9.U7.ACCA treats patients with scAAV9, which then uses host polymerases to generate functional U7snRNA, PMOs are synthesized ex vivo and provided to patients in their final functional form.

Currently, EST drugs exist to treat DMD patients amenable to the skipping of exon 45 (casimersen), exon 51 (eteplirsen), and exon 53 (golodirsen and viltolarsen) [15,16,17,18]. As was previously identified, all of these drugs target exons in the exon 43–55 hotspot, and no EST options are currently available or in clinical trials to treat deletions in the N-terminal hotspot. However, prior observations and analysis of the global Leiden DMD database have noted that patients harboring deletions of exons 3–7 or 3–9 often display relatively mild dystrophic phenotypes, in the latter case sometimes bordering on asymptomatic, suggesting that EST in this region could have great therapeutic potential for patients [11,60,61,62]. The mild phenotype associated with exon 3–7 deletions is particularly unusual, as this mutation is out-of-frame and therefore expected to result in a severe phenotype; however, patients with this deletion display significantly delayed age at loss of ambulation (LOA) [61]. The potential for EST in this region has been further supported by both in vitro and in vivo studies, as detailed below.

Kyrychenko et al. examined the effect of various multi-exon deletions in iPSC-derived cardiomyocytes containing a mutation in the N-terminal actin-binding domain (ABD-1) of *DMD* [63]. Moreover, iPSCs with an out-of-frame deletion of *DMD* exons 8 and 9 were treated with CRISPR/Cas9 to generate iPSC models with ΔEx3–9, ΔEx6–9, or ΔEx7–11. All three deletion patterns restore the reading frame of dystrophin and are hypothetical candidates for exon skipping therapy. IPSC-derived cardiomyocytes from the different models were then tested for function by analyzing dystrophin production, calcium handling ability, and contractile ability in engineered heart muscle (EHM). All three models demonstrated increased dystrophin production, improved regulation of calcium handling, and improved contractile ability compared to cardiomyocytes derived from the original ΔEx8–9 model. Notably, the ΔEx3–9 model showed the most pronounced functional recovery of the three in-frame models.

Based on this finding, Kyrychenko et al. further explored the function of exon 3–9 skipping in patient cells containing an out-of-frame ΔEx3–7. Patient-derived iPSCs were treated with CRISPR/Cas9 to generate ΔEx3–9 patient cardiomyocytes and assessed for dystrophin production and calcium handling once again. Similar to their previous study, they found that ΔEx3–9 cardiomyocytes displayed significantly improved calcium handling compared to ΔEx3–7 cells. More impressively, ΔEx3–9 cells showed a restored level of dystrophin production similar to healthy control cardiomyocytes. These findings ultimately demonstrate the applicability of exon skipping therapy to treat DMD caused by N-terminal mutations in the dystrophin gene and identify exon 3–9 skipping as a promising and effective therapeutic target.

Yokota et al. examined the in vivo safety and efficacy of exon 6–9 skipping in canine X-linked muscular dystrophy (CXMD) dogs [64]. ΔEx7 CXMD dogs were treated with intravenous injection of varying doses of a 3-AON cocktail designed to induce skipping of exons 6 and 8, which had been previously validated in vitro. All dogs showed evidence of systemic skeletal muscle dystrophin restoration, as well as cardiac dystrophin restoration to a lesser extent. Histological analysis revealed a decrease in centrally nucleated muscle fibers in treated dogs, as this is used as an indicator of muscular pathology, and MRI examination showed a decrease in T2 signal, as it is used to assess inflammation and water content. Finally, clinical assessments showed an improvement in 15 m run test performance for all treated dogs when compared to a decrease in performance for all untreated littermates. Additionally, evaluation of urea nitrogen, α-glutamyl transpeptidase, creatinine, amylase, total protein, total bilirubin, C-reactive protein, sodium, potassium, chloride, and body weight revealed no evidence of toxicology in dogs treated with the PMO cocktail, and there was no evidence of associated organ dysfunction or inflammation. Interestingly, Yokota et al. observed efficient skipping of exon 9 both in vitro and in vivo, despite the PMO cocktail only targeting exons 6 and 8. This suggests that spontaneous skipping of exon 9 is possible when using AONs to skip exon 8, which has important ramifications for minimizing the number of AONs required for multi-exon skipping. Overall, these data validate the in vivo therapeutic potential of N-terminal EST for DMD and provide a foundation for the development of future AON therapies in this region.

### Strengths and Weaknesses

As the only genetically targeted treatment to currently hold FDA approval, AON-mediated exon skipping therapy is leading the field of precision therapy for DMD, largely due to its favorable safety profile and demonstrated efficacy [65,66,67]. Approximately 30% of the DMD population is currently eligible for EST, using one of the four approved options, and this number is set to expand as research surrounding EST continues, providing a promising therapeutic option to an increasingly large subset of DMD patients [68,69]. Based on the positive results from Olsen et al. and Yokota et al., multi-exon-skipping therapy could be an attractive future option to address DMD arising from N-terminal dystrophin mutations. While details remain sparse, Astellas Gene Therapies has announced that it is working on a new AAV-antisense therapy candidate to treat mutations in exons 1–5 [70]. The drug, known as AT702, is in the preclinical stage and no further information has been provided at this point.

Despite its celebrated success, EST for DMD remains limited by a few main drawbacks. Due to the difficulties of PMO synthesis and requirement for recurrent treatments, EST can cost hundreds of thousands of dollars annually, and this could be a prohibitive factor for many patients based on local healthcare and insurance policies [16]. Furthermore, the regulatory approval for multi-exon skipping, such as the ΔEx3–9 therapeutic model identified by Olsen et al., is extremely difficult, as each individual AON in the cocktail must be evaluated in clinical trials independently [71]. This adds cost, time, and complexity to the development of AON cocktails for multi-exon skipping, delaying their availability to the patient population. Finally, exon skipping therapy suffers from poor tissue targeting, which may decrease its overall efficacy. Due to a variety of factors, such as quick renal excretion and their charge-neutral structure, less than 1% of AONs are estimated to actually reach the correct cellular compartment where they can have a therapeutic effect [23,72,73]. To address this limitation, recent studies have explored the effect of conjugating various peptides to PMOs to increase their cellular uptake [74]. Animal trials studying these peptide-conjugated PMOs (PPMOs) have shown improved exon-skipping efficiency in both skeletal and cardiac muscle compared to traditional PMOs; however, they also demonstrate increased nephrotoxicity associated with PPMOs [71,75]. Further studies continue to attempt to optimize this approach to ensure that it is both safe and effective for future clinical use [73].

## 4. Generalized Gene Therapy

Although it is not specific to any particular mutation pattern, the last form of precision therapy that has been gaining traction in the DMD community is known as microdystrophin therapy [41,76,77]. Microdystrophin therapy relies on the same basic underlying principle as exon skipping therapy: truncated dystrophin proteins display partial functionality and can impart some alleviation of the dystrophic phenotype. With this therapy modality, engineered dystrophin genes containing massive in-frame deletions are provided through a viral delivery vector, permitting host expression of exogenous truncated dystrophin. Studies exploring the minimum size for functional microdystrophin have identified progressively smaller dystrophin isoforms that are still functionally significant, and they have advanced to the point where microdystrophin can even be encoded within the stringent 4.7 kb limit of the AAV vector [76,77]. In vivo studies using the dystrophic dog model have identified that AAV-mediated delivery of canine microdystrophin is associated with increased dystrophin expression, improved muscle histological phenotype, and improvements in clinical parameters [41,78,79]. Based on these promising findings, and the success of a similar gene-replacement therapy aiming to treat spinal muscular atrophy, multiple commercial research groups have begun clinical trials exploring the therapeutic potential of AAV-delivered microdystrophin for the treatment of DMD [80]. Pfizer was the first to announce clinical trials for microdystrophin therapy with its candidate PF-06939926, but it was quickly followed by Sarepta Therapeutics (SRP-9001) and Solid Biosciences (SGT-001). Below is a summary of the current state of clinical trials for each candidate.

### 4.1. Pfizer-PF-06939926—Phase I

NCT03362502 is a phase-1 clinical trial titled “A phase 1b multicenter, open-label, single ascending dose study to evaluate the safety and tolerability of pf-06939926 in ambulatory and non-ambulatory subjects with Duchenne muscular dystrophy” [81]. As suggested by the official title, this is an open-label study primarily monitoring the safety of microdystrophin treatment in an estimated cohort of 35 DMD patients. Pfizer’s drug candidate, PF-06939926, uses a recombinant AAV9 vector to deliver microdystrophin regulated by a muscle-specific promoter. Included participants are males 4 years and older with a confirmed diagnosis of Duchenne muscular dystrophy and who have been previously treated with daily glucocorticoids for at least 3 months and have not attempted any other forms of gene therapy. Patients with antibodies against AAV9 are excluded from this study, and enrollment is still in progress. Participants will be treated with a single intravenous injection of either 1 × 10^14^ vg/kg (low-dose) or 3 × 10^14^ vg/kg (high-dose) PF-06939926 and monitored for treatment-related adverse effects over the course of 1 year. Secondary outcomes include analyzing dystrophin expression through immunohistochemistry and Western blot at the 2-month and 1-year marks, as well as longer-term safety studies monitoring clinical parameters such as adverse effects, body weight, vitals, and cardiac performance for 5 years following treatment.

Preliminary interim results from Pfizer’s press release were made available in May 2020 for nine patients treated with PF-06939926 [82]. These data indicated that vomiting, nausea, decreased appetite, and pyrexia were common adverse events occurring in >40% of participants. Furthermore, three serious adverse events requiring urgent intervention were observed in the first two weeks following treatment: persistent vomiting, resulting in dehydration; acute kidney injury with associated complement activation; and thrombocytopenia with associated complement activation. All three events were effectively treated and had resolved completely within 2 weeks. Exploratory endpoints from this interim data identified a significant increase in microdystrophin production and significant improvement in North Star Ambulatory Assessment (NSAA) score at the 12-month mark associated with PF-06939926. During a subsequent update at the Muscular Dystrophy Association’s (MDA) Scientific and Clinical Conference in March 2021, Pfizer announced that it had dosed another 10 patients with PF-06939926, for a total of 19 [83]. Notably, there were no more serious adverse advents presented at this update, which Pfizer attributed to the implementation of a glucocorticoid-based mitigation plan inspired by its previous interim results. Thirty percent of patients in the larger cohort experienced the previously mentioned minor adverse events, and improvements in NSAA scores were consistent with previous findings. Based on these promising findings, PF-06939926 received fast-track designation from the FDA to begin phase-3 trials in late 2020 (NCT04281485) [84].

However, concerning findings have since arisen regarding patient safety. In September 2021, three patients were reported by Pfizer to have experienced serious treatment-related muscle weakness, two of whom also experienced myocarditis, leading to the exclusion of patients who have mutations in exon 9 through 13 or both exon 29 and 30 from all trials [85]. Most recently, in December 2021, Pfizer reported the death of a patient being treated with PF-06939926 [86]. No further details have been provided by Pfizer at this point, and both NCT03362502 and recruitment for the upcoming phase 3 NCT04281485 have been put on hold pending investigation.

### 4.2. Pfizer-PF-06939926—Phase III

NCT04281485 is a randomized double-blind placebo-controlled study primarily meant to determine the efficacy of PF-06939926 [84]. Ninety-nine ambulatory male patients between the ages of 4 and 7 years with confirmed DMD will be recruited for this study. All patients must be on stable glucocorticoid therapy, and patients who test positive for AAV9 or who have mutations in either exons 9–13 or exons 29–30 will be excluded. Participants will be treated with a single intravenous infusion of either PF-06939926 or placebo, in a 2:1 ratio, and functional improvement will be assessed by using change in NSAA from baseline at the 52-week mark. Secondary clinical parameters, serum analysis, and measurement of microdystrophin production by using immunohistochemistry and liquid chromatography-mass spectrometry will also be assessed at this point. After one year, patients in the placebo arm will also be treated with PF-06939926, and all patients will be monitored for 5 years. Recruitment for this study is currently on hold following the tragic death of a patient in NCT03362502. Pending the resumption of recruitment, the estimated study endpoint is February 2028.

### 4.3. Sarepta-SRP-9001—Phase I

Sarepta’s SRP-9001, also known as rAAVrh74.MHCK7.micro-dystrophin, relies on AAVrh74 to deliver microdystrophin under the control of MHCK7, a muscle-specific promoter with improved cardiac expression, and is currently involved in three clinical trials [87,88,89]. NCT03375164 is an open-label phase 1/2 trial with two experimental cohorts primarily assessing the safety of SRP-9001, with secondary endpoints measuring motor improvement and microdystrophin expression. Cohort A represents participants from the original phase 1 study, while cohort B represents participants in the Phase 1/2 extension to the same study. Cohort A recruited six males (originally four for phase 1) between 3 months and 3 years of age, while cohort B recruited six males between the ages of 4 and 7 years. Participants in both cohorts are DMD patients with a confirmed mutation between exons 18 and 58 of the dystrophin gene who are otherwise healthy, who test negative for AAVrh74 and AAV8 antibodies, and who have not received any other forms of gene therapy. Furthermore, participants in cohort A must test below average on the Bayley-III motor score and have no previous treatment with corticosteroids, while participants in cohort B must test below average on the 100 Meter Timed Test and have received oral corticosteroids for at least 12 weeks prior to beginning the study. All patients received a single intravenous infusion of 2.0 × 10^14^ vg/kg SRP-9001, and analysis will focus on treatment-related adverse events over the course of 5 years post-injection. Secondary analysis will examine motor skills with the Bayley-III test (cohort A, over 3 years) or the 100 m Timed Test (cohort B, over 5 years), as well as microdystrophin gene expression via Western blot and immunofluorescence (both cohorts, 90 days post-treatment).

Data have been made available from the original 1-year study in four patients from cohort A, and demonstrated that no serious adverse events were associated with SRP-9001 treatment [87]. Eighteen treatment-related minor adverse events occurred, nine of which were vomiting, and three patients showed temporarily elevated γ-glutamyltransferase which resolved with corticosteroid treatment. All patients demonstrated an increase in microdystrophin production and an improvement in NSAA score. These promising preliminary results allowed for the extension of NCT03375164 and the beginning of phase 2 trials, NCT03769116.

### 4.4. Sarepta-SRP-9001—Phase II

NCT03769116 is a randomized double-blind placebo-controlled phase 2 trial assessing the efficacy of SRP-9001 in 41 males with DMD which began in December 2018 [89]. Inclusion criteria required that participants be males between the of ages 4 and 7 with genetically confirmed DMD and who have been taking oral corticosteroids for at least 12 weeks prior to the study and who are otherwise healthy. Patients were treated with single intravenous injection of either SRP-9001 or placebo, and analyzed after 48 weeks for a change in baseline NSAA score and after 12 weeks for a change in microdystrophin expression measured via Western blot. Secondary outcomes assessed other clinical benchmarks, such as time of 100 m timed test, time to rise from floor, and time to ascend four steps, as well as further quantifying microdystrophin using immunofluorescence. After 48 weeks, patients in the treatment group will be moved to the placebo group and vice versa, and all patients will be treated with intravenous injection once again. Analysis will reoccur 48 weeks after this second round of treatment, followed by a 3-year open-label extension for all patients.

In January 2021, Sarepta announced topline results via press release for the first 48-week period of NCT03769116 [90]. Patients treated with SRP-9001 showed significantly increased microdystrophin expression at 12 weeks compared to baseline, with a mean expression of 28.1%, and no new safety concerns arose. However, there was no significant improvement of NSAA score compared to placebo at the 48-week mark, which was the primary functional endpoint of the study. Sarepta claimed in its release that this discrepancy was due to poor randomization, which allocated patients with a higher baseline NSAA to the placebo group who thus had a better natural history, and it remained optimistic about results from the second phase of NCT03769116.

### 4.5. Sarepta-SRP-9001—ENDEAVOUR

In late 2019, prior to any updates on the efficacy of its phase-2 trials, Sarepta announced a major partnership with Roche for the continued development of SRP-9001, resulted in the commencement of yet another clinical trial, NCT04626674, better known as the ENDEAVOR trial [91,92]. Labeled as a phase-1 trial, ENDEAVOUR began in November 2020 and primarily aims to assess the change in microdystrophin expression in DMD patients treated with SRP-9001. Thirty-eight patients with confirmed DMD who are otherwise healthy and test negative for antibodies to AAVrh74 will be recruited and treated open-label with a single intravenous infusion of SRP-9001. Primary outcomes will use Western blot to measure the change in microdystrophin expression 12 weeks post-treatment compared to baseline, while secondary outcomes will assess adverse events, vector shedding, and the development of antibodies to AAVrh74. Immunofluorescence will also be used to assess microdystrophin localization.

Interim data from the first 11 patients in this study were presented at the World Muscle Society Virtual Congress in September 2021 [93]. These data demonstrated that SRP-9001 treatment was associated with robust microdystrophin expression which localized to the sarcolemma and also demonstrated that expression was associated with the presence of vector genome copies. Furthermore, the safety profile was determined to be consistent with that of previous studies, and all treatment-related adverse effects were temporary and able to be managed.

Shortly after these data were presented, Sarepta followed up with a press release providing interim clinical data from all three of its clinical trials: NCT03362502, NCT03769116, and the ENDEAVOR study [94]. Contrary to the previous failed topline results from NCT03769116, the new interim analysis of NCT03769116 showed a significant increase of 2.9 points (*p* < 0.0129) in NSAA scores for patients treated with SRP-9001 compared to matched natural history controls. Participants in NCT03362502 also showed a significant increase in NSAA score after 3 years of treatment, 8.6 points greater than their matched natural history cohort (*p* < 0.0001). Finally, an early analysis from the ENDEAVOR trial showed a 3.0-point improvement from baseline NSAA score after six months of SRP-9001 treatment. These positive results fueled the development of phase-3 trials for SRP-9001, NCT05096221 [95].

### 4.6. Sarepta-SRP-9001—EMBARK

NCT05096221, also known as EMBARK, is the phase-3 collaboration between Sarepta and Roche to finalize the development of SRP-9001 [95]. It is a randomized double-blind placebo-controlled trial designed to assess the efficacy of SRP-9001 treatment for patients with DMD. The study is currently in the early stages of recruiting 120 participants, and eligibility criteria require participants to be DMD patients between 4 and 7 years of age who are otherwise healthy, ambulatory, and do not show elevated AAVrh74 antibody titers. All participants must also be on oral corticosteroids for at least 12 weeks prior to trial commencement. Patients will be treated with a single intravenous transfusion of either SRP-9001 or placebo. Change in NSAA score from baseline will be assessed at the 52-week mark as a primary outcome. Secondary outcomes at this time point will explore a variety of other clinical function tests, and microdystrophin production will also be quantified with Western blot at 12 weeks post-treatment. At the 52-week point, patients in the SRP-9001 group will be treated with a placebo, and the placebo group will be treated with SRP-9001. NSAA and secondary outcomes will once again be recorded after 52 weeks. This study is estimated to complete in November 2024, and the results from this study will be critical in determining whether SRP-9001 receives FDA approval for the treatment of DMD.

### 4.7. Solid Biosciences-SGT-001—IGNITE

The final microdystrophin therapy candidate undergoing FDA review is Solid Biosciences’ SGT-001, which is currently in Phase 1/2 clinical trials (NCT03368742) [96]. SGT-001 uses the AAV9 vector to encode a functional microdystrophin product containing a unique neuronal nitric oxide synthase (nNOS) binding domain which is believed to aid in muscle protection from ischemic damage [97]. NCT03368742 is an open-label study designed to assess the safety and preliminary efficacy in DMD patients treated with SGT-001. An estimated 16 patients will be treated with single intravenous infusion of 5 × 10^13^ vg/kg (low-dose) or 2 × 10^14^ vg/kg (high-dose) SGT-001. Originally, the study was designed to include an untreated control group; however, it was removed after the first four participants were recruited. Inclusion criteria for the study require that participants are males between 4 and 17 years with genetically confirmed DMD who are otherwise healthy and who are not receiving any other forms of gene therapy. All participants must use oral corticosteroids for at least 12 weeks prior to beginning the trial, and they must be negative for the presence of AAV9 antibodies. Following treatment, microdystrophin quantification from at the 12-month mark will be used as the primary efficacy endpoint. Safety will also be assessed throughout by observing for adverse events, clinical, vital, and laboratory abnormalities. This trial is still actively recruiting patients to fill its quota and has an estimated completion date of December 2028.

This study was heavily delayed by two separate clinical holds in 2018 and 2019, which were imposed following the occurrence of serious treatment-related adverse effects, including complement activation, reduced platelet count, liver dysfunction, and acute kidney injury [98,99]. More recent interim data from the first six patients enrolled in the study were made available via press release from Solid Biosciences in March 2021 [100]. The release revealed that patients in the low-dose cohort experienced a mean increase in NSAA score of 1.0, while those in the high-dose cohort increased by 0.3. Patients from the untreated control declined by 4 points in the same timeframe, suggesting preliminary clinical benefit associated with SGT-001 treatment. Patients in both dose cohorts experienced clinically important improvements in the 6-min walk test, improvements in forced vital capacity, and meaningful improvements as assessed by Patient-Reported Outcome Measures (PROMs). Four serious treatment-related adverse effects arose in the first three patients treated, which prompted protocol amendments, and no serious adverse effects have been reported since. The original adverse events have since been fully resolved. Due to the low number of patients who have been tested and the difficulties which Solid has faced during this trial, no firm conclusions can be drawn from this study regarding the safety or preliminary efficacy at this point.

### 4.8. Strengths and Weaknesses

Compared to the mutation-specific tailoring of EST, microdystrophin offers a compelling treatment option that would be applicable to a much wider subset of DMD patients [76]. Especially for patients with rare mutations which might not be addressed by EST for many years, microdystrophin therapy could present the closest achievable gene therapy to treat DMD caused by mutations in the N-terminal region. With three main microdystrophin candidates in clinical trials—two of which are already in phase 3—microdystrophin is closer to market than any competing exon-skipping approaches for the N-terminal region.

However, it is important to note that the majority of data available regarding the performance of microdystrophin therapy in clinical trials thus far comes in the form of press releases from the sponsoring agencies that may not be as reliable or convey as comprehensive a picture as a true primary research publication and must, therefore, be interpreted with a degree of caution. No true conclusions regarding efficacy can be drawn until the completion of clinical trials. Furthermore, while the price of these drugs is not yet available, the cost of similar AAV therapies can be over $2 million for a single patient, due to the high costs of AAV production, which could put these therapies outside of the financial reach of many, if not most, patients [101]. Finally, while the size restriction associated with AAV therapy was cleverly circumvented through the development of microdystrophin, the other limitations of AAV therapy discussed previously are still present with the AAV-mediated delivery of microdystrophin. Namely, interactions with the host immune system and the possible safety risks of AAV could prove a major limitation, especially given the plethora of adverse events which occurred in both the early and late stages of Pfizer’s and Solid Bioscience’s trials, as well as the recent and tragic death of one of Pfizer’s participants. Although protocol modifications and the concurrent use of glucocorticoid therapy appear to help control these adverse effects, longer-term safety data must still be generated to properly validate these drugs. A brief summary of all clinical trials listed for microdystrophin therapy can be found in Table 1.

## 5. Conclusions

A variety of drug candidates and gene-therapy modalities are in development that have the potential to treat some or all of the members of the DMD population containing mutations in the N-terminal hotspot of the dystrophin gene. Due to the complexity of the therapies and the rapid rate at which new discoveries are made in the field, it is extremely difficult to reliably predict which options might or might not meet with success. However, based on the multitude of serious adverse events and recent deaths associated with AAV-based therapies for both DMD and X-linked myotubular myopathy, approaches relying on AAV for their delivery will potentially face delays in approval and more stringent safety validation down the road. These additional difficulties could make other approaches, such as EST, comparatively more attractive for groups aiming to bring DMD precision therapy to market, shifting the focus away from the ever-popular AAV. As previously mentioned, approval for multi-exon cocktail EST currently faces its own regulatory hurdles as each AON must be evaluated separately by the FDA. In the event that the safety profile of AAV-based therapies proves unsuitable for DMD, it could prompt discussions with the FDA regarding changes in its policy for cocktail approval in order to address a current unmet need in the DMD patient population. Alternatively, future efforts might focus on different vectors for microdystrophin and CRISPR delivery. A successful microdystrophin therapy could provide a broad treatment option with applicability to more patients than a single AON or cocktail could achieve, but development is being bottlenecked by AAV-related difficulties. If vectors with comparable efficacy and an improved safety profile can be identified for microdystrophin and CRISPR delivery, it could help to circumvent this issue. Regardless of their outcomes, each clinical or preclinical trial mentioned has and will continue to improve our understanding of DMD and brings the DMD community one step closer to improved precision-therapy options.

## Figures and Tables

**Figure 1 genes-13-00257-f001:**
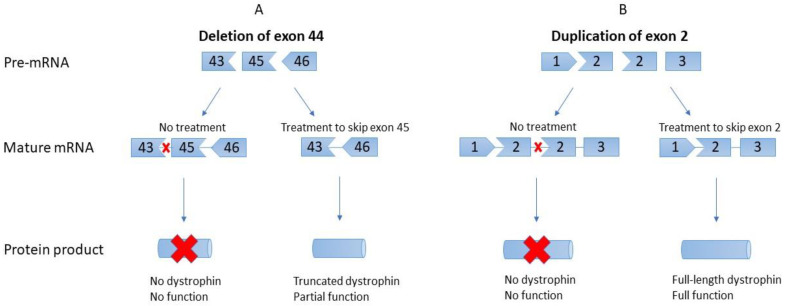
Differences in exon skipping for deletions compared to duplications. (**A**). Deletions, such as the pictured deletion of exon 44, require skipping of an adjacent exon to restore proper reading frame, in this case exon 45. The resulting product is an internally deleted but functional dystrophin product. (**B**). In contrast, duplications require the direct skipping of the duplicated exon, resulting in the return of normal, full-length dystrophin with normal function. In this case, the duplicate of exon 2 is skipped to restore the healthy transcript.

**Table 1 genes-13-00257-t001:** A summary of clinical trials currently underway for microdystrophin therapy indicating phase, sponsoring agency, study type, endpoints, and end date.

Trial	Drug Candidate	Phase	Study Type	Primary Endpoint	Estimated or Actual Primary Completion Date
NCT03362502	PF-06939926 (Pfizer)	1	Open-label dose escalation study	Adverse events	February 2022
NCT04281485	PF-06939926 (Pfizer)	3	Randomized double-blind placebo-controlled study	Clinical efficacy with NSAA score	February 2023
NCT04626674 (ENDEAVOR)	SRP-9001 (Sarepta)	1	Open-label efficacy study	Change in microdystrophin expression	March 2022
NCT03375164	SRP-9001 (Sarepta)	1/2	Open label safety study	Adverse events	April 2023
NCT03769116	SRP-9001 (Sarepta)	2	Randomized double-blind placebo-controlled study	Clinical efficacy with NSAA score and change in microdystrophin expression	December 2020
NCT05096221 (EMBARK)	SRP-9001 (Sarepta)	3	Randomized double-blind placebo-controlled study	Clinical efficacy with NSAA score	October 2023
NCT03368742	SGT-001 (Solid Biosciences)	1/2	Open-label dose escalation study for safety and efficacy	Adverse events, change in microdystrophin expression, and clinical abnormalities	December 2023

## Data Availability

Not applicable.

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
