# Peer review of "Antisense and Gene Therapy Options for Duchenne Muscular Dystrophy Arising from Mutations in the N-Terminal Hotspot"

_genes, 2022, doi:10.3390/genes13020257_

Round 1
Reviewer 1 Report
This review entitled « Antisense and gene therapy options for Duchenne muscular dystrophy arising from mutations in the N-terminal hotspot” by Harry Wilton-Clark and Toshifumi Yokota, summarizes perfectly the current state of development of different therapies for DMD, focusing on mutations in the N-terminal region of dystrophin.
The Astellas gene therapies consist on AAV delivery vectors. Here the aim of the scAAV9-U7-ACCA is to treat DMD patients with a duplication of exon 2. They aim to induce the skipping of exon 2.
Even though skipping both exon 2 produces a truncated but functional protein, I suggest adding a sentence to explain how they go about inducing single exon 2 skipping.
The authors have clearly stated the absence of off target risk and toxicity. They have exposed the advantages and disadvantages of this therapy. An important point is the part describing the risk of immune reaction due to AAV in 30% to 60% of the population. They propose a validation of the safety of AAV before this type a clinical trial to ensure patient health.
In the same concern to clearly explain the principle, the advantages and the disadvantages, they summarize the work carried out on therapy by PMOs. The use of PMO to remove an exon is well known and several teams are now proposing to remove several exons. They underline the limit of this therapy: difficulty in synthesizing PMOs, the need to treat recurrently, costs and the need to specifically develop the AONs cocktail for each patient!
Microdystrophin is the third type of therapy discussed in this article which involves the delivery of a functional truncated version of dystrophin by AAV. Three different trials are exposed with two already in phase 3. This approach could offer a treatment option applicable to a wide subset of DMD patients.
Finally, I do not have a very great expertise on DMD and current therapies but I find that this review is very well written, it is objective with systematically positive points but also limits for each trials
I am totally in favor of publishing this review without criticism. I am convinced that it provides a good overview of what is currently being done in this area!
Author Response
Reviewer 1: Even though skipping both exon 2 produces a truncated but functional protein, I suggest adding a sentence to explain how they go about inducing single exon 2 skipping.
Author response: Thank you for the suggestion. We have included further information to increase the clarity of this point and explain why no additional measures are in place to skip a single exon 2. It now reads:
“Astellas has previously demonstrated that the treatment of mice containing a duplication of exon 2 with scAAV9.U7.ACCA can induce skipping of exon 2 and result in amelioration of the DMD phenotype [28]. As the U7snRNA is unable to differentiate between the identical duplicates of exon 2, there exists the possibility of skipping both copies. However, Astellas found that even when there is complete exclusion of both copies of exon 2, production of a truncated but functional protein is maintained through alternative translation initiation at an internal ribosome entry site (IRES) on exon 5 in both human cells and mice [29]. This finding suggests that the therapeutic window for exon skipping in exon 2 duplication patients is larger than previously expected, and indicates that no additional measures are required to favor single-exon exclusion.”
Reviewer 2 Report
This is a very nice, complete review. My expertise is in the field of oligonucleotide therapeutics. I do not have the expertise to judge the prodigious amount of information specific to Duchenne Therapy or gene therapy as a strategy. I trust that the other reviewer(s) will make up for my deficit.
1) line 53: ASOs aren't gene therapy. Two approaches are being used, gene therapy using vectors and synthetic oligonucleotides. They are fundamentally different. These differences need to be described. In particular, the strengths and weaknesses of the two approaches. While this paper focuses on gene therapy, some context relative to ASOs would help.
2) As the paper progresses, the paragraphs get longer and longer. I appreciate the plethora of facts, but they paper becomes a brutal read. Paragraphs should be broken up into shorter, better focused subunits.
3) The authors are obviously experts in this area. Despite that, I don't detect much expert opinion or perspective. They have put in a massive effort. I urge them to provide expert opinion as well as straight facts. The conclusion is especially weak. While I appreciate that the field is complex, the authors must have opinions that they would be excited to share. As long as opinions are marked as opinions, I can only see them as value added.
4) As noted above, the procession of facts tends to be a bit brutal and that will be only somewhat alleviated by a reorganization of paragraphs. Additional figures summarizing data/clinical design might help.
In summary, I thank the authors for this outstanding summation and recognize the effort behind the work. I found it valuable and the authors have done a service for the community. I hope that accommodating my comments will be simple and yield a final product that will be of more lasting value to a broad readership.
Author Response
Reviewer 2: line 53: ASOs aren't gene therapy. Two approaches are being used, gene therapy using vectors and synthetic oligonucleotides. They are fundamentally different. These differences need to be described. In particular, the strengths and weaknesses of the two approaches. While this paper focuses on gene therapy, some context relative to ASOs would help.
Author response: Thank you for bringing this up. Changes have been made throughout the paper to better highlight this distinction. A section was also added in the introduction to briefly differentiate the two. It reads:
“Precision therapies for DMD include gene therapy and antisense therapy. Gene therapy tends to be used as an umbrella term for a variety of more specific approaches, and while the exact definition varies widely, it usually refers to therapies that rely on the production of recombinant genetic materials, such as microdystrophin and CRISPR/Cas9, provided to cells [19–21]. Gene therapy has the potential to provide long-lasting therapy with a single treatment, effectively treating some diseases at their root. However, unwanted mutations, immunogenicity, and off-target effects can pose a safety risk to patients, limiting the applicability of some methods of gene therapy [21]. In comparison, antisense therapy directly treats patients with synthetically produced DNA-like molecules known as antisense oligonucleotides (AONs) targeted for mRNA without providing new genes to express, in the same fashion as many other drugs might be provided to a patient [22,23]. While AONs generally have a favorable safety profile, the requirement for repeated treatments can add cost and complexity, especially in the setting of chronic disease [16].”
Reviewer 2: As the paper progresses, the paragraphs get longer and longer. I appreciate the plethora of facts, but they paper becomes a brutal read. Paragraphs should be broken up into shorter, better focused subunits.
Author response: Long sections were further separated to improve clarity, especially in the clinical trials section of exon skipping therapy which had particularly long paragraphs.
Reviewer 2:The authors are obviously experts in this area. Despite that, I don't detect much expert opinion or perspective. They have put in a massive effort. I urge them to provide expert opinion as well as straight facts. The conclusion is especially weak. While I appreciate that the field is complex, the authors must have opinions that they would be excited to share. As long as opinions are marked as opinions, I can only see them as value added.
Author response: The conclusion was expanded to provide better details and the author’s opinion opinion. It now reads:
“A variety of drug candidates and gene therapy modalities are in development which have the potential to treat some or all of the DMD population containing mutations in the N-terminal hotspot of the dystrophin gene. Due to the complexity of the therapies and the rapid rate at which new discoveries are made in the field, it is extremely difficult to reliably predict which options might or might not meet with success. However, based on the multitude of serious adverse events and recent deaths associated with AAV-based therapies for both DMD and X-linked myotubular myopathy, approaches relying on AAV for their delivery will potentially face delays in approval and more stringent safety validation down the road. These additional difficulties could make other approaches such as EST comparatively more attractive for groups aiming to bring DMD precision therapy to market, shifting the focus away from the ever-popular AAV. As previously mentioned, approval for multi-exon cocktail EST currently faces its own regulatory hurdles as each AON must be evaluated separately by the FDA. In the event that the safety profile of AAV-based therapies proves unsuitable for DMD, it could prompt discussions with the FDA regarding changes in their policy for cocktail approval in order to address a current unmet need in the DMD patient population. Alternatively, future efforts might focus on different vectors for microdystrophin and CRISPR delivery. A successful microdystrophin therapy could provide a broad treatment option with applicability to more patients than a single AON or cocktail could achieve, but development is being bottlenecked by AAV-related difficulties. If vectors with comparable efficacy and an improved safety profile can be identified for microdystrophin and CRISPR delivery, it could help to circumvent this issue. Regardless of their outcomes, each clinical or pre-clinical trial mentioned has and will continue to improve our understanding of DMD, and brings the DMD community one step closer to improved precision therapy options.”
Reviewer 2: As noted above, the procession of facts tends to be a bit brutal and that will be only somewhat alleviated by a reorganization of paragraphs. Additional figures summarizing data/clinical design might help.
Author response: In addition to changing paragraphs to be shorter and better focused (as per point #2), improved headings were added to the clinical trials section of EST to better organize the procession of facts and break up long walls of text. Additional figures were explored but determined to add more clutter than value.